# Recent Development in Detection and Control of Psychrotrophic Bacteria in Dairy Production: Ensuring Milk Quality

**DOI:** 10.3390/foods13182908

**Published:** 2024-09-13

**Authors:** Kidane Yalew, Xiaoyang Pang, Shixin Huang, Shuwen Zhang, Xianchao Yang, Ning Xie, Yunna Wang, Jiaping Lv, Xu Li

**Affiliations:** 1Key Laboratory of Agro-Products Quality and Safety Control in Storage and Transport Process, Ministry of Agriculture and Rural Affairs, Institute of Food Science and Technology, Chinese Academy of Agricultural Sciences, Beijing 100193, China; 2018y90100121@caas.cn (K.Y.); pangxiaoyang@163.com (X.P.); zswmaster@163.com (S.Z.); xiening@caas.cn (N.X.); wang_yn92@163.com (Y.W.); kjdairy@126.com (J.L.); 2Department of Vet. Public Health and Food Safety, College of Veterinary Sciences, Mekelle University, Mekelle 0231, Tigrai, Ethiopia; 3Shanghai Animal Disease Control Center, No. 30,855 Nong, Hongjing Rd., Shanghai 201103, China; huangshx1968@163.com (S.H.); yangxianchaoml@163.com (X.Y.)

**Keywords:** milk spoilage, psychrotrophic bacteria, heat-resistant enzyme, detection methods, control strategies, cold storage

## Abstract

Milk is an ideal environment for the growth of microorganisms, especially psychrotrophic bacteria, which can survive under cold conditions and produce heat-resistant enzymes. Psychrotrophic bacteria create the great problem of spoiling milk quality and safety. Several ways that milk might get contaminated by psychrotrophic bacteria include animal health, cowshed hygiene, water quality, feeding strategy, as well as milk collection, processing, etc. Maintaining the quality of raw milk is critically essential in dairy processing, and the dairy sector is still affected by the premature milk deterioration of market-processed products. This review focused on the recent detection and control strategies of psychrotrophic bacteria and emphasizes the significance of advanced sensing methods for early detection. It highlights the ongoing challenges in the dairy industry caused by these microorganisms and discusses future perspectives in enhancing milk quality through innovative rapid detection methods and stringent processing controls. This review advocates for a shift towards more sophisticated on-farm detection technologies and improved control practices to prevent spoilage and economic losses in the dairy sector.

## 1. Introduction

Due to its high nutritional content, raw milk can serve as a perfect environment for various microorganisms to grow. It can be easily contaminated, most notably with spoilage bacteria, during handling, storage, transportation, and processing [1]. In the dairy industry, storing raw milk in refrigerators for less than 10 °C before processing for 2 to 5 days is common practice to slow down the growth of pathogenic mesophilic and thermophilic bacteria. Still, this practice also creates ideal conditions for the development of psychrophilic (cold-loving) and psychrotrophic bacteria, which can continue to grow at low temperatures (≤7 °C) and eventually dominate the milk microflora [2].

Most psychrotrophic bacteria found in milk can produce hydrolytic thermostable enzymes that break down milk’s major constituents: milk fat, protein, and lecithin [3]. It has been well established by independent research groups in different countries that *Pseudomonas* spp. is an essential psychrotroph that dominates the microbiota of raw milk during cold storage. This genus of Gram-negative, aerobic, and rod-shaped bacteria harbors species with a well-established physiological mechanism of adaptation and growth at low temperatures, with the shortest generation times from 0 to 7 °C [4]. Among the known genera, *Pseudomonas*, *P. fluorescens*, *P. fragi*, and *P. putida* are the most common species, and they are recognized as producers of extracellular thermostable enzymes, mainly proteolytic and lipolytic. These bacterial enzymes have been the most studied and best described [5]. Though these bacteria can be destroyed by pasteurization or ultrahigh temperature (UHT), the thermotolerant hydrolytic enzymes (protease and lipases) produced during milk storage and transportation under refrigeration can withstand thermal treatment used during the manufacture of the majority of dairy products. However, several of these enzymes can keep their activity between 60 and 70% after pasteurization and 30 and 40% after the sterilization of milk [6]. In general, these residual activities of enzymes, present at low concentrations, that degrade milk content can alter the physicochemical properties of the processed milk over time [7], causing technological challenges for the dairy sector economy by limiting the shelf life and quality of processed milk and dairy products [1]. It is estimated that psychrotrophic bacteria can lead to up to 30% in production losses in the dairy industry due to their enzymatic spoilage activities, significantly impacting financial outcomes [8,9]. 

Controlling the proliferation of spoilage bacteria during refrigerated raw milk storage is crucial to maintaining the shelf life of the derived dairy product. For a better understanding of the dairy microbiota, numerous techniques have been applied to explore the diversity of bacteria in raw milk. Traditional phenotypic methods for bacterial identification are based on the isolation and growth of microorganisms on solid media, followed by the observation of morphological and biochemical characteristics [10]. Those approaches are usually inexpensive and straightforward, but they can entail laborious culture procedures, be time-consuming, and often be insufficient for identifying bacterial strains. Various techniques rely on the isolation and growth of pure bacterial strains, which is problematic because some bacteria are non-culturable [11,12].

Recently, the use of molecular identification methods has circumvented this obstacle. Rapid molecular methods with high sensitivity and specificity have been developed to overcome the limitations of conventional approaches to detect spoilage microorganisms in raw milk [13]. The advancement in rapid detection methods for spoilage microorganisms in milk and the scientific community’s contribution towards developing time-saving, specific, fast, and efficient methods of detection are underlined. Therefore, this review provides a comprehensive review of psychrotrophic bacteria, their characteristics, heat-stable enzymes, detection methods, control strategies, and future perspectives based on recent publications that could help ensure milk quality.

## 2. The Role of Psychrotrophic Bacterial Challenges in Milk Quality

Dairy products and milk provide optimal conditions for the growth of a broad spectrum of microorganisms, primarily due to their high nutritional content, near-neutral pH, and substantial water content [14,15,16]. Bacterial spoilage in food represents a significant global challenge, impacting the economic stability of the food industry due to inadequate processing and refrigeration facilities [4]. To maintain consumer loyalty, it is imperative for dairy industries worldwide to ensure the high quality and extended shelf life of their products [2]. The microbial diversity in unprocessed milk is complex. It varies based on numerous factors, including the cattle’s health, personnel hygiene, feed type, mammary gland condition, milking equipment, surrounding air, bedding, and water, as well as the regional climate and seasonal changes (Figure 1) [10]. Although various factors contribute to milk spoilage during the milking process, the predominant microbiota is primarily influenced by the conditions under which raw milk is stored and transported, as well as the duration of storage before processing [17].

While refrigeration is widely used to extend the shelf life of raw milk and inhibit spoilage by mesophilic bacteria, it also alters the bacterial community in a way that favors the growth of psychrotrophs as well as psychrophiles. These bacteria, capable of producing heat-resistant enzymes, are commonly found in water, soil, vegetation, and occasionally in the air [18]. Since the introduction of bulk refrigerated storage, psychrotrophic bacteria have become predominant in raw milk microbiota, even under sanitary conditions. Post-refrigeration, their numbers can increase from about 10% of total mesophilic aerobes to an average of 90% [19]. Under unsanitary conditions, psychrotrophic bacteria, which thrive at freezing temperatures, can constitute up to 75% of the original microbiota in raw milk. This widespread distribution indicates a high degree of genetic and physiological adaptability, including the production of cold shock proteins, changes in membrane composition to maintain fluidity, and the production of antifreeze proteins, enabling them to thrive in harsh and cold environments [20].

In healthy udder cells, milk is typically sterile. It has low bacterial concentrations upon leaving the udder, generally meeting European Commission (EC) standards for milk quality as measured in colony-forming units [18]. Levels of psychrotrophic bacteria exceeding 1.3 × 10^5^ CFU/mL are considered significant enough to compromise raw milk quality spoilage. Concentrations of this microbial community in unprocessed raw milk vary with location and season, typically ranging from 10^2^ to 10^7^ CFU/mL. The microbiological quality of drinking and cleaning water, animal diet, and the hygiene of milking equipment and operations are all closely linked to the levels of microbial contamination in raw milk [2,11]. These bacteria play important roles in nutrient cycling in cold environments, the decomposition of organic matter in polar regions, and as part of microbial communities in permafrost soils. In addition, they are significantly studied in the food industry as they can grow slowly at refrigeration temperatures, leading to the spoilage of refrigerated foods such as dairy products, meat, and seafood [21].

## 3. The Impact of Heat-Resistant Enzymes and Their Technological Challenges

The extracellular enzymes produced in milk by psychrotrophic bacteria, including proteases, lipases, phospholipases, exopeptidases, and glycosidases, are recognized for their heat stability. Proteases and lipases, in particular, are well understood due to their enhanced hydrolytic characteristics [6,9,22]. The dairy industry faces technological challenges due to hydrolytic enzymes produced by psychrotrophic bacteria during cold storage. Although high-temperature treatments eliminate microorganisms, the thermoresistant enzymes produced by the microorganisms can withstand the thermal processes commonly employed in dairy production during milk transportation and refrigerated storage [9,23,24,25]. Yet, these hydrolytic enzymes can be beneficial in the dairy industry, as they contribute to the development of flavor and texture in cheese during the ripening process. However, these heat-resistant hydrolytic enzymes also pose technological challenges in milk and dairy products, as they remain active throughout product processing [5,26,27]. Some of the genera and strains of psychrotrophic bacteria with significant proteolytic and lipolytic activity are listed (Table 1).

Predominantly, *Pseudomonas*, a genus isolated from cold raw milk, is known for secreting heat-resistant hydrolytic enzymes, according to various studies [34]. The predominance of *Pseudomonas* is consistent across most sampling locations, regardless of the isolation and identification methods used or the duration of milk storage. The different literature revealed that *Pseudomonas* is the main genus associated with the breakdown of key milk components, with *P. fluorescens* being the most common species responsible for reducing the shelf life of raw milk and processed dairy products through spoilage, significantly reducing their economic value. In addition to this, the enzymatic activity of these bacteria leads to the breakdown of proteins and fats, causing off-flavors, bitterness, and textural changes, which are undesirable characteristics in dairy products [24,35]. In cheese production, psychrotrophic bacteria and their enzymes result in reduced yield and tainting problems through the degradation of casein into peptides and amino acids by proteases, resulting in these components being lost into the whey rather than forming the curd, thereby decreasing the cheese yield. Tainting is primarily caused by proteolytic and lipolytic activities producing bitter peptides and free fatty acids, resulting in strong and often undesirable flavors. In addition to spoilage, as mentioned earlier, blue discoloration in fresh mozzarella cheese was greatly reported and drew the attention of many researchers in the dairy industry, who named its blue mozzarella [19,36].

Another significant issue is the “age gelation” phenomenon in UHT milk, which is characterized by an increase in viscosity and the eventual formation of a gel, leading to a loss of fluidity. Studies have indicated that the psychrotrophic bacterial population of 5.5 log CFU/mL in raw milk can cause UHT milk gelation after 20 weeks of storage. In contrast, higher populations can induce this defect in a shorter period, between 2 to 10 weeks [37,38]. Psychrotrophic bacteria can also interfere with the activity of starter cultures used in dairy fermentation processes. The presence of proteases and lipases may alter the growth rates and activity of these cultures, consequently affecting the final product’s fermentation process and quality, leading to increased rennet coagulation times and deteriorated texture in fermented dairy products such as cheese and yogurt [11]. Biofilm formation by psychrotrophic bacteria on dairy processing equipment poses a persistent contamination risk. These biofilms consist of bacterial communities embedded in a self-produced extracellular polymeric substance (EPS) matrix, making them difficult to eradicate. They can act as a continuous source of bacterial contamination, leading to repeated spoilage issues in subsequent batches of milk [39]. Though defects are challenges for processing high-quality milk by different dairy industries, scientists are still doing their best to achieve their target in the production of safe and high-quality dairy products by advancing their rapid detection methods.

## 4. Factors Affecting Enzyme Production in Psychrotrophic Bacteria

Enzyme production by psychrotrophic bacteria in dairy processing industries is affected by different factors, among which the incubation temperature, storage time, strain specificity, bacterial community, and quorum sensing (QS) are some of them, as reported by Jimenez et al. [40]. The quality of dairy products is determined by the enzymatic production capacity of the different genera, species, and strains of psychrotrophic bacteria [32]. Spoilage by bacterial species in milk is primarily driven by cell density-dependent signaling molecules released by Gram-negative bacteria, such as *P. fluorescens*, as well as by Gram-positive bacteria [41,42]. At high cell densities, bacteria communicate with each other using chemical signaling molecules in a process known as quorum sensing [43,44].

The primary chemical signaling molecules involved in QS include acylated homoserine lactone (AHL) in Gram-negative bacteria, autoinducing peptides (AIPs) using modified oligopeptides in Gram-positive bacteria, and interspecies autoinducer-2 (AI-2) hybrid systems in both bacterial types, aiding researchers in understanding biological phenomena such as biofilm formation and pollutant degradation across various bacterial species [45,46]. However, these signaling molecules are degraded by an enzymatic process known as quorum quenching (QQ), which is known for its role in inhibiting biofilm formation and suppressing the production of virulence factors [47,48]. QQ is defined as the enzymatic degradation of quorum-sensing signaling molecules. This reduction in signaling molecule concentration due to QQ enzymes in milk storage tanks and processing lines inhibits biofilm formation by psychrotrophic bacteria, thereby reducing the risk of microbial contamination load. Consequently, QQ negatively impacts bacterial enzyme production through the enzymatic breakdown of signaling molecules involved in quorum sensing [48,49].

The temperature during bacterial incubation, pasteurization, and UHT treatment significantly affects enzymatic activity, as demonstrated by the marked differences in enzyme synthesis in psychrotrophic bacteria incubated at 30 °C compared to 7 °C [50]. Seasonal variations also have a significant impact on enzymatic activity; as per a report by Dai et al. [51], samples collected in autumn showed higher spoilage rates compared to those collected in summer from different provinces of China, because the high temperatures in summer inhibit the growth of psychrotrophic bacteria by mesophilic bacteria as compared to the favorable autumn temperatures. While samples collected in winter also had more spoilage potential as compared to samples collected in summer, this was more related to cross-contamination from inadequate hygiene practices related to animal bedding, water, and sanitation [9,19].

## 5. Detection Methods of Psychrotrophic Bacteria

Modern techniques for quick milk spoilage detection and identification are becoming more and more necessary due to the wide range of spoilage microorganisms that affect milk quality in the dairy industry and other food sectors [12,52]. Numerous detection methods have been employed to gain a comprehensive understanding of the milk microbiota and to explore the diversity of psychrotrophic bacteria in raw milk [53]. In addition to this, these recent advanced sensing methods utilize biomarkers, which possess unique physical and chemical properties that provide specificity, speed, and efficiency in detecting microbial spoilage through targeted hybridization and amplification techniques (as shown in Figure 2), such as using nucleic acids, proteins, macromolecules, and metabolic products [2,54,55].

Some of the recent advanced sensing (Figure 2) methods currently used for detecting foodborne microbial spoilage in milk include the following: (1) nucleic acid base methods like conventional polymerase chain reaction (PCR), quantitative PCR (qPCR), multiplex PCR (mPCR), droplet digital PCR (ddPCR), and other methods such as DNA microarray, isothermal amplification, which is loop-mediated isothermal amplification (LAMP), recombinase polymerase amplification (RPA), recombinase-aided amplification (RAA), fluorescent in situ hybridization (FISH), and sequencing (Sanger sequencing, next-generation sequencing (NGS), and third-generation sequencing (TGS)) [55]; (2) biosensor-based methods such as optical, piezoelectric, immunosensor, and electrochemical biosensors; (3) immunological methods like enzyme-linked immunosorbent assay (ELISA), lateral flow immunoassay, immunofluorescence assay (IFA), serum neutralization tests (SNTs), and immunomagnetic separation assay [56]; and (4) mass spectrometry methods such as matrix-assisted laser desorption ionization time-of-flight mass spectrometry (MALDI-TOF MS) [57].

### 5.1. Nucleic Acid Base Detection

#### 5.1.1. Polymerase Chain Reaction (PCR)

PCR is a widely utilized molecular method for detecting foodborne bacterial pathogens [29]. To evaluate the detection of spoilage microorganisms by PCR, strains are inoculated separately into a selected broth and were subjected to DNA extraction after 18 to 24 h of incubation. The process involves three main steps: initially, the double-stranded target DNA is denatured at high temperatures to form single-strand DNA (SSD) [13]. Originally, the detection limit for PCR in pure milk culture would identify positive results when the concentration of *P. fluorescens* ranged from 10^7^ to 10^9^ CFU/mL. However, advancements in molecular techniques and improvements in filtration methods have enhanced the sensitivity of PCR, now detecting as few as 10^4^ CFU/mL [58]. This highlights PCR is a time-saving and highly sensitive method, surpassing the culturing method in various aspects. However, it has its limitations, among which is its inability to differentiate between live and dead cells. This lack of discrimination can lead to excessive control measures and economic losses within the dairy industry. Furthermore, the requirement for thermocycling to separate DNA strands restricts its usability in resource-limited settings [16].

#### 5.1.2. Real-Time PCR (RT-PCR)

Unlike conventional PCR, real-time PCR, also known as quantitative PCR (qPCR), eliminates the need for agarose gel electrophoresis to analyze PCR results. Instead, it utilizes the fluorescence intensity that correlates directly with the quantity of PCR amplicons produced [59,60,61,62]. This method is extensively used for detecting and quantifying microorganisms across various research domains [60,63]. Research has shown that TaqMan-based qPCR offers greater sensitivity compared to SYBR Green I or molecular beacon-based approaches [12]. In recent applications within food microbiology, qPCR has been extensively adopted for detecting psychrotrophic bacteria in milk. According to a study by Wang et al. [64], the minimal detection limit of *P. aeruginosa* targeting the gene UCBPP-PA14 by qPCR was identified with 10^2^ CFU/mL, which was significantly more sensitive than endpoint PCR and traditional culturing methods, enhancing sensitivity by one to two orders of magnitude (Table 2). Consequently, Martinez et al. [65], from their comparison of the traditional culture method vs. the qPCR method, clearly defined that the time taken from pre-enrichment, enrichment, and up to detection for microorganisms was around 6 days and the material required was greater as compared to the qPCR method. For this reason, qPCR demonstrates greater specificity, sensitivity, and efficiency in microbial detection than the traditional way and has significant value, proving to be a promising technique for the rapid identification of milk spoilage microorganisms.

#### 5.1.3. Loop-Mediated Isothermal Amplification (LAMP) Detection

Over the past two decades, numerous innovative isothermal nucleic acid amplification techniques have been developed to suit low-resource settings and point-of-need applications [66]. Among these, loop-mediated isothermal amplification (LAMP) stands out as a significant advancement since its inception by Notomi [67]. LAMP operates through auto-cycling strand displacement DNA synthesis at isothermal temperatures ranging between 59 and 65 °C, utilizing a reaction mixture of nucleotides, Bst DNA polymerase, primer sets, and reaction buffer containing magnesium ion. The technique employs four sets of primers with outer forward (F3), inner forward (FIP), outer backward (B3), and inner backward (BIP) primers to target six specific regions of DNA, enhancing by the addition of a loop primer pair, which hybridizes facilitating the generation of stem-loop DNAs of various sizes and complex, cauliflower-like DNA structures with multiple loops [68]. The LAMP assay has demonstrated its capability to detect a broad spectrum of pathogens, ranging from simple *Escherichia coli* to the latest SARS-CoV-2 [69], which can produce a significantly higher volume of amplicons within 60 min, often more than 10^3^ times that of traditional PCR. This capacity makes LAMP notably more sensitive and faster, with lower detection limits for identifying foodborne pathogens than standard PCR methods [70].

This method has gained significant popularity due to its rapid and cost-effective nature in detecting various food pathogens. The optimized LAMP assay exhibited a lower detection limit compared to a conventional PCR-based method. In pure culture, the LAMP assay demonstrated a detection limit of 4.8 × 10^1^ CFU/reaction of template DNA, whereas the PCR method presented a detection limit of 4.8 × 10^2^ CFU/reaction. The evaluation of method performance in *P. fluorescens*-contaminated pasteurized cow milk revealed the detection limit of real-time LAMP and PCR assay to be 7.4 × 10^1^ CFU/reaction and 7.4 × 10^3^ CFU/reaction, respectively, where the detection limits of the real-time LAMP and PCR assays were performing using 7.4 × 10^5^ to 7.4 × 10^−1^ CFU/reaction of the template DNA, which was two-fold times lower than that of the PCR-based method (Table 2) [71,72,73]. With further development, the LAMP assay has the potential to provide a favorable on-farm alternative to existing technologies for the detection of psychotropic bacterial contamination in milk, thereby enhancing the quality control of milk and milk products [71,72]. Therefore, this is to highlight that LAMP assays align with the World Health Organization (WHO) Affordable, Sensitive, Specific, User-friendly, Rapid, Equipment-free, and Deliverable (ASSURED) criteria, making it an ideal diagnostic tool for use in resource-limited environments [70,74].

#### 5.1.4. Recombinase Polymerase Amplification (RPA) Detection

In recent years, with the continuous development of nucleic acid-based amplification technology, RPA stands out as a relatively straightforward approach. Commercialized by TwistDx (www.twistdx.co.uk (accessed on 21 May 2024)), RPA utilizes proteins involved in cellular DNA synthesis, recombination, and repair. This method of isothermal amplification has gained significant attention due to its simplicity, rapid performance, and wide applicability with regards to its fast response with time, ability to tolerate specific mismatches, uncomplicated primer design, and facilitation of multiplex amplification reactions [66,75].

The RPA method is highly sensitive and selective, capable of amplifying 1–10 target DNA copies within ≤20 min at a range of 37–42 °C [76,77]. It has shown excellent compatibility with multiplexing and is known for its rapid amplification capabilities, which have been successfully used to amplify a variety of targets, including RNA, miRNA, ssDNA, and dsDNA from multiple samples and organisms [78]. Thus, through *Mycoplasma bovis* amplification targeting the amplicon *uvrC* gene dsDNA, using a direct RPA assay and the rapid detection of the bacterium in bovine milk, it was determined that the limit of detection was 1.0 × 10^1^ copies per reaction. This is more sensitive to the observed limit of detection in the endpoint PCR and real-time PCR assay [79,80,81,82].

However, RPA also has its limitations, among which the high cost incurred by the kit is one of them, as the kits are available exclusively from one company. In addition, there is no dedicated software for designing RPA-specific primers, potentially complicating the primer design process for sequence specificity and sensitivity. Moreover, RPA is primarily designated for research purposes as it has not yet received Food and Drug Administration (FDA) approval. Despite these challenges, it is a new standard set for nucleic acid amplification in the foreseeable future, with further research likely to enhance its capabilities and applications [66,75,83]. A comparative summary for the detection limit sensitivity of each nucleic acid base detection for the spoilage milk microbes is listed in Table 2.

**Table 2 foods-13-02908-t002:** Milk microbial spoilage detection limits based on nucleic acid detection method applications.

Microorganism Detection	Target Gene	Sample Type	Limit Detection	Reference
**Endpoint PCR**				
*P. fluorescens*	*aprX*	Milk	10^4^ CFU/mL	[58]
*S. aureus*	*UCBPP-PA14*	Milk	10^4^ CFU/mL	[64]
**qPCR**				
*P. aeruginosa*	*UCBPP-PA14*	Milk	10^2^ CFU/mL	[64]
*B. cereus*	*gyrB*	Milk	2 × 10^2^ CFU/mL	[84]
**LAMP**				
*P. fluorescens*	*lipA*	Pasteurized milk	7.4 × 10^1^ CFU/mL	[73]
*P. fluorescens*	*aprX*	Pasteurized milk	3 × 10^2^ CFU/mL	[71]
**RPA**				
*M. bovis*	*uvrC*	Milk	10^1^ CFU/mL	[79]
*E. coli O157:H7*	fliC, stx and rfbE	Milk and water	10^1^ CFU/mL	[82]

#### 5.1.5. Biosensor-Based Method Detection

Compared to traditional detection technology, advanced methods such as test strips and biosensors offer technological innovation. They are efficient, quick, and convenient for the detection of foodborne pathogens in both food and the environment [85]. A biosensor is an analytical device consisting of two primary components: a bio-receptor and a transducer. The bio-receptor is tasked with recognizing specific targets, which include enzymes, antibodies, proteins, nucleic acids, aptamers, and cell receptors. Upon recognition, the transducer component converts biological interactions into measurable electrical signals. This conversion process is foundational for various types of biosensors, including optical, piezoelectric, immunosensor, and electrochemical biosensors [86,87]. Biosensors are renowned for providing rapid and real-time detection, capable of monitoring multiple bacterial targets simultaneously directly at the site of interest [88]. Among them, optical biosensors stand out for their selectivity and sensitivity, making them exceptionally suitable for the real-time monitoring of toxins, drugs, and pathogens [55].

Electrochemical biosensors, which detect foodborne pathogens through potentiometry, conductometry, and impedimetry, have become widely used in the fields of food, biology, and life sciences due to their numerous advantages [85]. These advantages include rapid processes, high sensitivity, high specificity, low cost, portability, miniaturization, and point-of-care detection. By providing a fast and efficient alternative method for detecting foodborne pathogens, electrochemical biosensors contribute to ensuring the safety of ready-to-eat (RTE) foods. They can also serve as standalone devices for on-site monitoring [89].

According to a study conducted by Alexandre et al., an amperometric biosensor was used to detect *Salmonella* Typhimurium in milk. The study demonstrated the biosensor’s specificity by testing it with pure and mixed samples containing strains of *E. coli* and *Citrobacter freundii*. The biosensor performance was satisfactory in detecting *Salmonella* Typhimurium quickly in both skim and whole milk samples without the need for an enrichment step. The biosensor had a very low limit of detection, 10 CFU/mL, and a detection time of 125 min. This immunosensor assembly can be further explored in future studies to detect other bacterial species in different food matrices, making it a valuable tool for ensuring food safety [90].

#### 5.1.6. Immunological Base Method Detection

Immunological detection represents a cornerstone method for identifying food pathogens in the food industry, utilizing antigen–antibody binding techniques. These methods include enzyme-linked immunosorbent assay (ELISA), lateral flow immunoassay, and immunomagnetic separation assay. Immunoassay has been used in pasteurized milk and raw milk with a low detection limit of 10^4^ to 10^6^ cells/mL within 30 min to 16 h [91]. Central to these assays is the specific interaction between an antibody and its corresponding antigen [92]. Various antibodies are utilized across different assays to detect foodborne pathogens and microbial toxins [93].

The effectiveness of these antigen–antibody complexes largely depends on the specificity of the antibodies used. Monoclonal antibodies, which provide a consistent source of a single type of antibody, are particularly valuable for the specific detection of target molecules and are often preferred over polyclonal antibodies. The advent of monoclonal antibodies has enhanced the specificity, sensitivity, reproducibility, and reliability of immunological detection. Consequently, numerous commercial immunological assays now reliably detect a broad range of microorganism and their byproducts [56]. Despite their specificity and capability for multiplexing across multiple samples, these methods are occasionally limited by false-negative results and potential cross-reactions with similar antigens [55].

#### 5.1.7. Mass Spectrometry Method Detection

Mass spectrometry (MS) is a technique utilized for the identification of microorganisms and bacteria by detecting their mass-to-charge ratio (*m/z*). It is a robust analytical tool capable of simultaneously detecting multiple targets [57]. Matrix-assisted laser desorption/ionization time-of-flight mass spectrometry (MALDI-TOF-MS) is one MS method and is a highly effective technique employed for the identification and differentiation of various microorganisms, including bacteria, viruses, parasites, and fungi [94]. This method utilizes a laser to ionize molecules within a sample, and the resulting ions are subsequently analyzed based on their mass-to-charge (*m/z*) ratios using a time-of-flight detector [95]. The process commences with the sample being mixed with an energy-absorbing matrix, which facilitates the desorption and ionization of analytes upon laser beam exposure. The ionized molecules are then accelerated in an electric field within the TOF analyzer, where they undergo separation based on their *m/z* ratios, ultimately generating a unique mass spectrum specific to the microorganism [96]. The identification of microorganisms through MALDI-TOF-MS involves comparing the obtained mass spectra to an extensive internal database of known spectra. This comparison necessitates intricate algorithms that match the experimental spectra to reference spectra, enabling identification at the genus and species levels. The utility of MALDI-TOF-MS extends beyond clinical diagnostics, finding applications in environmental monitoring, food safety, and biodefense. The technology has been utilized for the identification of microbial contaminants in water and food, as well as potential bioterrorism agents such as *Bacillus anthracis* and *Yersinia pestis* [95]. The method offers advantages due to its ability to handle complex samples with minimal preparation and the presence of highly conserved protein biomarkers that ensure accurate detection. Notably, MALDI-TOF-MS boasts expedited processing time, with comparative studies demonstrating that results can be achieved in as quick as 30 min, significantly faster than traditional methods requiring 24 to 48 h. Despite these advantages, MALDI-TOF-MS does face challenges, such as the potential for misidentification caused by database limitations and the inherent similarity between certain species [96]. However, ongoing updates to spectral databases and improved algorithms are addressing these concerns, enhancing the accuracy and reliability of microbial identifications. The future of MALDI-TOF-MS in microbial detection involves its potential integration with genomic and proteomic data to further enhance its identification accuracy and ability to directly detect antimicrobial resistance and virulence factors from clinical specimens [96]. These advancements are expected to result in more comprehensive and rapid diagnostics, broadening the range of applications for MALDI-TOF-MS in microbial research and clinical practice. A summary of the detection methods with their advantages and disadvantages is listed in Table 3.

## 6. Control Strategies of Psychrotrophic Bacteria in Milk

Ensuring that raw milk is obtained under sanitary conditions is imperative to reduce the initial contamination by psychrotrophic bacteria. This involves cleaner cows, the better sanitary design of equipment, enclosed pipeline milk systems, and more rigorous cleaning methods. The quality of dairy products is significantly impacted by the presence of microorganisms and their thermostable enzymes, leading to substantial global economic losses due to product deterioration. Raw milk can be contaminated from various sources, making it challenging to control bacterial entry and subsequent enzyme production in unprocessed milk [8,97]. Moreover, a comprehensive understanding of the entire microbiotic diversity in fresh milk is essential to prevent bacterial contamination during the storage, transportation, and milking processes. The dairy industry frequently contends with issues caused by psychrotrophic bacteria and their heat-resistant enzymes. Even a low bacterial count, such as 1.3 × 10^5^ CFU/mL before processing, can lead to technological problems, including milk gelation, formation, and yield losses in cheese production and other milk products [23,50]. Several strategies can be employed, some of which are discussed below, from the time of milking at the farm level to the final packaging process to limit the growth of spoilage-causing psychrotrophic bacteria [33].

### 6.1. Temperature

Various temperature treatments are applied to control and reduce the levels of psychrotrophic bacteria and their enzymes, which compromise the quality of unprocessed milk. The rapid pre-cooling and pre-thermalization of fresh milk are crucial since the temperature of milk at udder excretion is approximately 35 °C, an ideal condition for microbial growth [98]. The combination of pre-heating for a specific duration (such as 72 °C for 15 s) followed by cooling to 2 °C (plus or minus 1 °C) has proven effective in maintaining the quality of stored milk for 2 to 5 days before further processing. This approach significantly enhances the quality of the final dairy product [8]. Otherwise, storing warm milk directly from the udder could dramatically increase the temperature within silos, fostering bacterial growth and enzymatic activity. The other critical point we should consider is the time before the start of cooling not exceeding 1 h, as in [99].

### 6.2. Modification of Atmospheric Packaging

Another effective technique for preserving milk and inhibiting microbial growth is modified atmospheric packaging, which includes the addition of food additives and flushing with nitrogen and carbon dioxide. This method is particularly significant in limiting the growth of psychrotrophic bacteria [100]. By replacing oxygen with nitrogen in packaging, oxygen-induced rancidity is prevented, and the growth of aerobic microorganisms is inhibited, leading to a reduction in microbial growth in unprocessed milk.

Modifying the atmospheric composition surrounding the milk, particularly through flushing with nitrogen gas and storing at low temperatures, has shown a strong inhibitory effect on bacterial growth [63]. Additionally, the use of carbon dioxide in treating raw milk has proven effective in reducing microbial activity. This approach has been thoroughly researched and demonstrates substantial potential for restricting microbial growth and enzyme production [18]. When raw milk is flushed with pure nitrogen gas, there is a significant reduction in bacterial growth, particularly of *Pseudomonas*, and a decrease in lipolysis and proteolysis during cold storage, creating an environment unfavorable for anaerobes. Moreover, the introduction of carbon dioxide after storing raw milk for 120 days not only reduces lipolysis and proteolysis but also preserves the microbiological and physicochemical properties of the milk. This treatment facilitates the production of ultrahigh-temperature (UHT) milk with an extended shelf life and reduced proteolysis [101].

### 6.3. Hygiene

Dairy farms encounter multiple sources of contamination, which can adversely affect milk quality. Water serves as a predominant source of microbial contamination, requiring stringent monitoring to ensure bacteriological quality during production. Other significant contamination sources include spoiled silage, improperly cleaned milking equipment, and fecal matter from dairy cows [102]. The means of milking significantly impacts the level of contamination in dairy farms. Proper udder hygiene and the use of disinfectants are critical during milking to minimize bacterial growth. Techniques such as foremilk stripping into clean cups can help detect abnormalities, including mastitis, thereby reducing the risk of cross-contamination. However, contaminated milking equipment and inadequate hygiene practices can lead to elevated bacterial counts in milk [103]. Psychrotrophic bacteria and their enzymes have the potential to cause defects in dairy products. To prevent these defects, various measures have been taken to limit bacterial growth and the formation of biofilm [104].

These measures include effective cleaning and sanitation practices, vital in reducing psychrotrophic bacteria, which compromise milk quality. The regular sanitization of milking equipment, storage tanks, and surfaces is essential for preventing bacterial contamination. Best practices include flushing equipment with lukewarm water to remove milk debris, conducting hot alkaline washes to eliminate bacteria, and finishing with warm acid rinses to lower pH, which discourages bacterial growth [103]. Adhering to a comprehensive cleaning routine and implementing rigorous quality control protocols that include continuous monitoring and sampling can significantly mitigate the risks associated with spoilage in milk production [105]. Some advanced dairy farm management systems, such as good hygiene practice (GHP) and good manufacturing practice (GMP), are also employed to minimize the initial number of microorganisms in unpasteurized milk. However, the impact of these sophisticated management systems on thermoresistant enzymes is still unknown [106]. Additionally, the formation of biofilm on stainless steel surfaces by milk microorganisms presents a hygiene challenge in milk processing. Efforts are being made to address this issue through the use of different chemicals and detergents [107].

### 6.4. Physical Preservation

High-pressure processing (HPP) is a food preservation technique that employs extreme pressure to destroy microorganisms in food. For milk, HPP is advantageous because it enables the inactivation of spoilage organisms and pathogens, guaranteeing food safety while extending shelf life. This method stands out as it maintains the organoleptic and nutritional properties of milk, which can be compromised by traditional thermal pasteurization methods [108]. HPP operates by applying isostatic pressure to milk within its final packaging. This process typically operates at pressures between 300 and 700 MPa, which helps to inactivate a wide range of pathogenic and spoilage microorganisms without causing significant thermal damage. The high-pressure environment disrupts cellular structures within bacteria, leading to the loss of viability while preserving the non-covalent bonds integral to milk’s sensory properties [109]. The primary benefit of HPP in milk processing is that it can extend the shelf life of milk products by weeks or even months, reducing the frequency of spoilage. It can also preserve nutritional quality unlike thermal pasteurization, whereby HPP retains more nutrients and bioactive compounds, such as immunoglobulins and lactoferrin, which are critical for health [110]. Having minimal sensor changes, HPP preserves the taste, color, and texture of milk, making it a preferred method for many producers [108]. Despite its potential, the application of HPP in the dairy industry faces significant hurdles. Regulatory frameworks in various regions, including stringent definitions of “raw” milk, can limit marketing possibilities for HPP-treated products. For instance, in the European Union, HPP is often equated with heat pasteurization, complicating how HPP milk can be labeled.

Additionally, concerns over the recovery of injured bacteria after HPP treatment necessitate ongoing research to ensure the complete safety of HPP milk. HPP presents a promising alternative to conventional pasteurization, significantly contributing to food safety and the preservation of milk quality [111]. Continued innovation and refinement of HPP techniques could alleviate current regulatory burdens and improve adoption in the dairy industry. Furthermore, research into the long-term effects of HPP on various milk components will be essential for optimizing its application [112]. HPP stands as a transformative technology, offering the potential to redefine milk preservation practices while maintaining quality and safety.

### 6.5. Natural Preservatives

Natural preservatives derived from milk, such as enzymes and specific compounds, significantly extend the shelf life of dairy products. They exert their effects through various mechanisms, primarily by inhibiting microbial growth, and have practical applications in food preservation processes [113]. The growing consumer demand for safer and cleaner food products has prompted an increased focus on natural preservatives in the dairy industry. Natural preservatives in milk include various organic acids, proteins, and enzymes, contributing to its antimicrobial properties. Specific compounds such as lactoperoxidase, lactoferrin, and antimicrobial peptides (AMPs) are derived from milk and play an essential role in inhibiting microbial growth [114]. The most notable chemical preservatives used in dairy products are potassium sorbate, sodium benzoate, natamycin, and nisin, which are recognized for their efficacy in preventing spoilage while maintaining product quality [115].

Their mechanism of action for milk-derived natural preservatives is primarily based on their ability to inhibit microbial growth through various pathways. For instance, lactoperoxidase catalyzes the oxidation of substrates in the presence of hydrogen peroxide, creating inhibitory conditions for bacteria. Meanwhile, lactoferrin binds to iron, depriving bacteria of the essential nutrients they require for growth. Moreover, AMPs disrupt microbial cell membranes, leading to cell lysis and death. Collectively, these mechanisms ensure food safety and prolong the shelf life of dairy products [115]. When we look into its practical applications of natural preservatives in milk, it includes fermentation processes that enhance microbial stability and the use of cultured dextrose and cultured skim milk to inhibit mold growth in dairy products. For example, natamycin is commonly used in cheese production due to its effectiveness against molds and yeasts. Natural preservatives like these not only contribute to food safety but also meet consumer demands for clean labels and sustainable practices [116,117]. They help reduce waste and enhance the nutritional profile of dairy products by incorporating beneficial probiotics [116].

### 6.6. Microbial Enzymes

Lactase oxidase (LO) is a microbially derived enzyme naturally present in milk used to inhibit spoilage microorganisms in raw milk with antimicrobial properties. It oxidizes lactose into lactobionic acid and reduces oxygen, generating H_2_O_2,_ which activates the lactoperoxidase system (LPS) [118,119,120]. In a report by Flynn et al. [118], 19 L of pilot raw milk polluted with a *Pseudomonas* cocktail was treated with a concentration of 0.24 g/L LO for 3 days, then processed at UHT, resulting in monitoring for gelation. Ultimately, there was a significant difference in the particle size between the LO-treated milk and the control, observed as early as one month after processing, and gelation was not detected in LO-treated samples after 6 months of storage. Thus, using enzymes to inhibit thermostability-producing psychrotrophic bacteria will ensure milk quality and reduce post-production losses in the shelf-stable milk market sector [118]. A summary illustration of microbial milk control is shown in Figure 3.

## 7. Conclusions and Future Perspectives

In light of the evolving landscape of microbial detection technologies, safeguarding the quality of dairy products against psychrotrophic bacterial contamination and the spoilage they cause remains imperative. With psychrotrophic bacteria’s ability to produce heat-stable enzymes posing a significant threat to dairy economics and product integrity, future research must be directed towards the early detection and comprehensive characterization of these bacteria and their enzymatic profiles at the farm level. This proactive approach will help to mitigate economic losses and enhance the safety and quality of milk products. Implementing stringent hygiene practices remains the cornerstone of preventing contamination. Additionally, exploring innovative packaging solutions, such as nano-antibacterial materials, could offer new avenues for effectively controlling bacterial growth. Adopting such advanced methodologies beyond traditional culture methods will be vital in saving time, improving specificity and efficiency, and moving towards the goal of high-quality and safe dairy product production.

## Figures and Tables

**Figure 1 foods-13-02908-f001:**
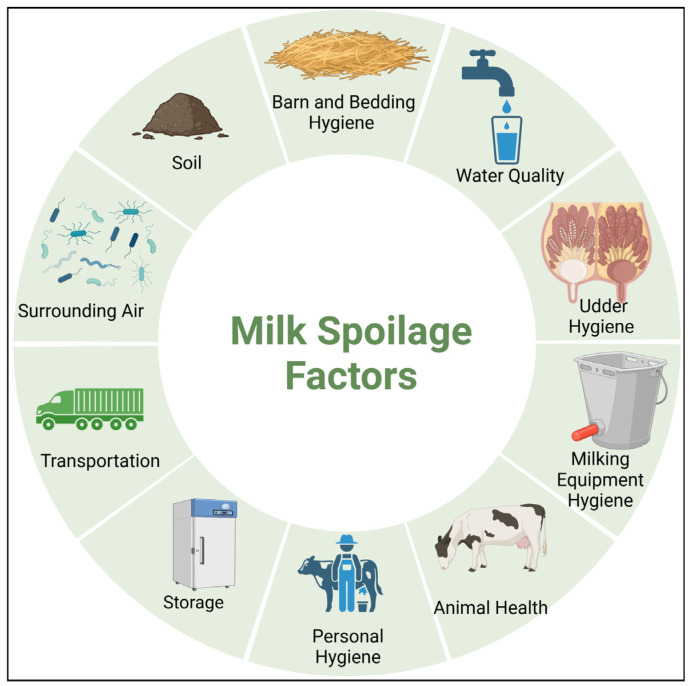
Diverse factors contributing to microbial milk spoilage.

**Figure 2 foods-13-02908-f002:**
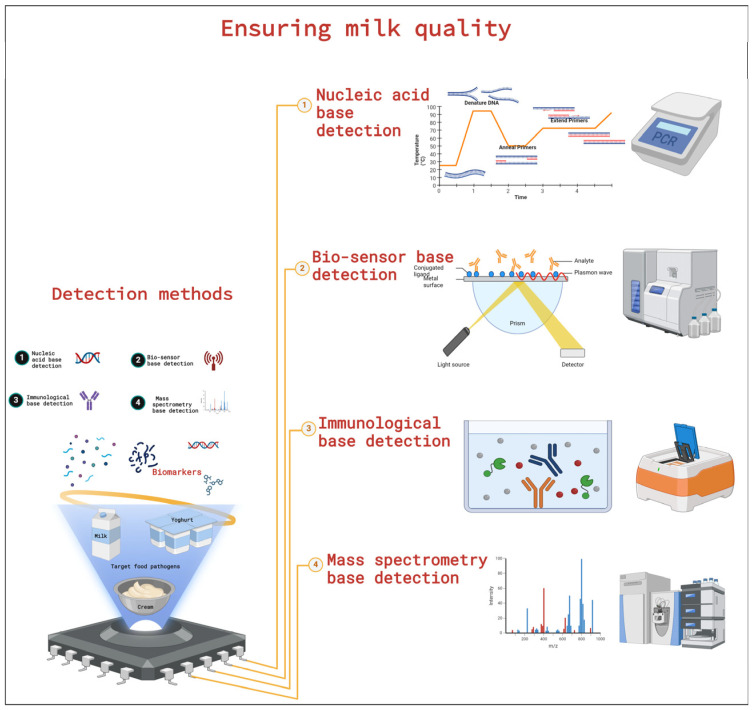
Detecting methods of milk quality through an integrated recent rapid sensing detection biomarker base.

**Figure 3 foods-13-02908-f003:**
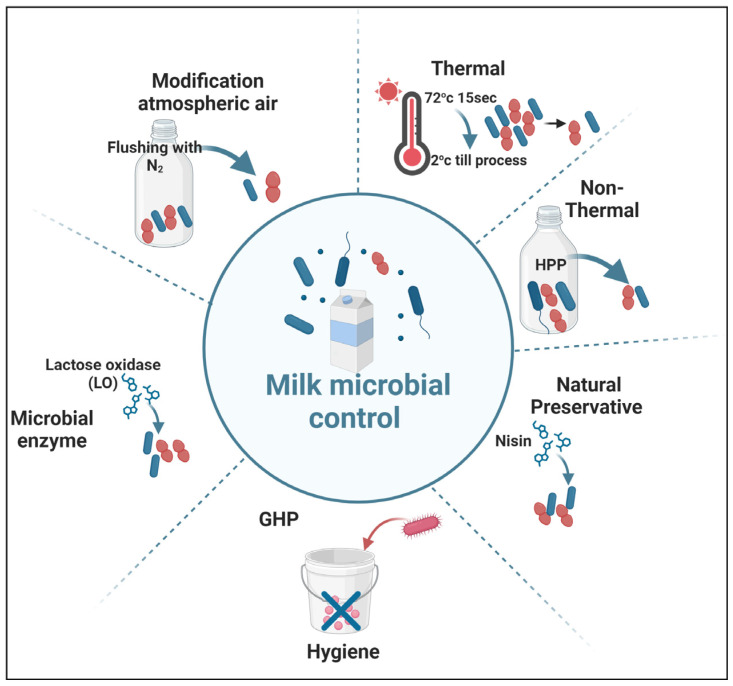
Summary of the diverse control methods for ensuring milk quality and limiting milk spoilage microbes. GHP, good hygiene practice; HPP, high-pressure processing.

**Table 1 foods-13-02908-t001:** List of psychrotrophic bacteria producing extracellular hydrolytic enzymes (lipase/protease).

Enzyme	Genus	Species	References
*Protease*	*Pseudomonas*	*Pseudomonas fluorescens* *Pseudomonas putida* *Pseudomonas veronii* *Pseudomonas fragi* *Pseudomonas lundensis* *Pseudomonas proteolytica*	[11,27,28,29,30,31]
	*Bacillus*	*Bacillus cereus*	
	*Serratia*	*Serratia liquefaciens* *Serratia grimesii*	
	*Hafnia*	*Hafnia alvei*	
	*Chryseobacterium*	*Chryseobacterium piscium* *Chryseobacterium oncorhynchi* *Chryseobacterium jejuense*	
	*Yersinia*	*Yersinia intermedia*	
*Lipase*	*Pseudomonas*	*Pseudomonas fluorescens* *Pseudomonas proteolytica* *Pseudomonas fragi* *Pseudomonas lundensis*	[11,27,28,29,30,31,32,33]
	*Bacillus*	*Bacillus* *c* *ereus*	
	*Acinetobacter*	*Acinetobacter guillouiae* *Acinetobacter johnsonii*	
	*Serratia*	*Serratia liquefaciens*	
	*Chryseobacterium*	*Chryseobacterium joostei* *Chryseobacterium scophthalmum*	
	*Lactococcus*	*Lactococcus raffinolactis*	
	*Enterobacter*	*Enterobacter kobei*	

**Table 3 foods-13-02908-t003:** Summary of the advantages and disadvantages of detection methods.

	Detection Method	Biomarker	Advantages	Disadvantages
1	Traditional culture-dependent	–	- Considered as the “gold standard” for many applications - Ability to detect a single bacterial strain - Recognition of viable cells - Appropriate for selective media	- Material-consuming - Time-consuming, often requiring several days to obtain results - They may also fail to detect viable but non-culturable bacteria - Risk of contamination
2	Advanced detection culture-independent	DNA/RNA, protein and enzymes	- Time-saving - Material-saving - Can detect viable but non-culturable bacteria - Multiple target detection and quantification	- They often require specialized equipment and expertise - More expensive - Less accessible in remote areas
	A. Nucleic acid base detection method	DNA/RNA	- High specificity - High sensitivity - Fast community profiling	- Require sample storage and processing - Require DNA/RNA extraction, which can cause RNA/DNA loss - Sensitive to inhibitors - High cost for large number of samples - Usually need specialized instruments
	B. Immunology-based methods	Protein	- Cost-effective - Can be automated - Can detect bacterial toxins	- Require pre-enrichment - Low sensitivity - Require labeling of antibodies and antigens
	C. Biosensor-based methods	DNA/RNA, protein, and chemicals	- High sensitivity - Real-time detection - Label-free	- High cost - Require specialized instrument - Low specificity - Not suitable for simultaneous detection of virous organisms - Low reproducibility and insufficient stability
	D. Mass spectrometry (MALDI-TOF)	DNA, protein, and macromolecules	- Fast detection time of 30 min - Cost-effective	- Misidentification of data limitation - Inherent similarity of species

## Data Availability

Not available.

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
