# Peer review of "Recent Development in Detection and Control of Psychrotrophic Bacteria in Dairy Production: Ensuring Milk Quality"

_foods, 2024, doi:10.3390/foods13182908_

Round 1
Reviewer 1 Report
Comments and Suggestions for Authors
The review submitted by Yalew et al. aimed to describe the importance, mechanisms of action, detection methods, and control of psychrotrophic microorganisms in the milk and dairy product production chain.
The topic is extremely important as it represents the main cause of economic losses for the dairy sector. The review is interesting and well-crafted, but in several sections, it addresses the content in a shallow and superficial manner.
The authors need to delve deeper into the topics since some consist of only one or two paragraphs and do not provide the current and comprehensive information on the topics discussed.
Furthermore, the authors should prioritize the inclusion of more recent manuscripts on the subject in this review. There are very few citations of articles published after 2020, with a predominance of references prior to this date (although there are some subsequent ones). The review should be as current as possible and requires a review and update of the included references.
Remarks:
- I suggest removing “safety” from the title. Psychrotrophic microorganisms are more related to quality loss rather than safety directly, despite some rare pathogens being psychrotrophic (e.g., L. monocytogenes). Moreover, the focus of the manuscript was not on safety.
- L17: Highlight the possible contamination during milk collection on farms, not just during transport and storage.
- In the Keywords, replace words already present in the title with others that contextualize the study area for better indexing.
- L29-35: Clarify that refrigeration, although allowing the multiplication of psychrotrophs, prevents the transmission of pathogens, which are mostly mesophilic microorganisms.
- L57: Do the authors know of any work that shows economic/financial loss? It would be interesting to include monetary values, if available.
- All figures mentioned in the manuscript were neither included nor found as a supplementary file for the review. The authors should include all of them with their respective captions immediately after their mention in the text. Provide complete and self-explanatory captions that allow understanding independent of the manuscript reading.
- L72-75: The authors should provide more detailed information about the combination of search terms for the articles. Which databases were consulted? What combinations of these words were used? What were the inclusion and exclusion criteria? If they do not have this information, I suggest removing it from the manuscript.
- I suggest removing the term "Results." To be considered a true result, a systematic search following guidelines such as PRISMA should have been conducted. As this was not done, it is a descriptive review and not systematic, so it is better to remove this term from the manuscript.
- The authors should review and standardize the numbering of the manuscript's topics and subtopics, which are inconsistent.
- L92: Conditions of the mammary gland as a whole, not just the teats.
- L100: Psychrotrophs as well as psychrophiles.
- L101: Reference the sources of contamination.
- In topic 1, I suggest including the following manuscript, which reports a technological problem not addressed in the review (https://doi.org/10.1016/j.idairyj.2021.105020).
- L145: Spores can survive, so it cannot be considered true sterilization, only commercial.
- In topic 1, more details about the technological defects caused by the action of psychrotrophic enzymes should be described.
- L176: Standardize the writing of the word Gram starting with a capital letter.
- L205, 215, 219, and other mentions in the manuscript: Spoilage instead of pathogen. The focus of the manuscript is on spoilage microorganisms, not pathogenic ones.
- In the PCR topic, specify that the prior isolation of the microorganism in culture media is necessary.
- L344, 346: Typhimurium should be written without italics and starting with a capital letter, as it is a serovar, not a species of the Salmonella genus.
- L372: Replace the term microbes with microorganisms throughout the manuscript.
- L391: Specify what "several days" would be. Be precise.
- In topic 5.1, the authors need to clarify which temperature control strategy should be used. It was difficult to extract practical and applicable information from the topic.
- L397: Wouldn't it be more applicable to solid products, like cheeses and butter?
- In topic 5.3 on hygiene, the authors should include information on contamination sources in dairy farms and how milking and cleaning should be performed to reduce psychrotrophic counts.
- In the topic on natural preservatives, the authors should provide more details and substances, with their mechanisms of action and practical applications.
- L447: Do not abbreviate Pseudomonas as P.
- Adjust the Credit authorship according to the journal's standards.
Author Response
A point-by-point response to reviewers
reviewer #1:
Comments and Suggestions for Authors
The review submitted by Yalew et al. aimed to describe the importance, mechanisms of action, detection methods, and control of psychrotrophic microorganisms in the milk and dairy product production chain. The topic is extremely important as it represents the main cause of economic losses for the dairy sector.
The review is interesting and well-crafted, but in several sections, it addresses the content in a shallow and superficial manner. The authors need to delve deeper into the topics since some consist of only one or two paragraphs and do not provide the current and comprehensive information on the topics discussed. Furthermore, the authors should prioritize the inclusion of more recent manuscripts on the subject in this review. There are very few citations of articles published after 2020, with a predominance of references prior to this date (although there are some subsequent ones). The review should be as current as possible and requires a review and update of the included references.
Author response to reviewer #1
Thank you for your careful and thorough reading of this manuscript and for the comments and constructive suggestions, which help to improve the quality of the manuscript. The manuscript has now been modified having considered all comments and suggestions raised by the reviewers. A point-by-point response to the reviewers is provided below.
Comments 1: I suggest removing “safety” from the title
Response : L3: Thank you for your professional advice. Yes, per your comment, we have removed the word “safety” from the title.
Comment 2: Line 17: Highlight the possible contamination during milk collection on the farm, not just during transport and storage.
Response : L18: Thanks. Yes, we accept your comment and we highlight the possible contaminations on the farm level during milk collection.
Comment 3: In the Keywords, replace words already present in the title with others that contextualize the study area for better indexing.
Response : L28: Thanks for your critical review. Per your comment, we replaced the words with others that contextualize the study area.
Comment 4: L29-35: Clarify that refrigeration, although allowing the multiplication of psychrotrophs, prevents the transmission of pathogens, which are mostly mesophilic microorganisms.
Response: L30-37: Thanks. We have rewritten, as per your comment, why and how refrigerator allow multiply of psychrotrophic and prevent transmission of the mesophilic microorganism.
Comment 5: L57: Do the authors know of any work that shows economic/financial loss? It would be interesting to include monetary values, if available.
Response: L56-59: We agree with your comment, which will be more interesting. We have been searching for such financial loss with monetary value, which is our interest, but we couldn’t get available articles for now.
Comment 6: All figures mentioned in the manuscript were neither included nor found as a supplementary file for the review. The authors should include all of them with their respective captions immediately after their mention in the text. Provide complete and self-explanatory captions that allow understanding independent of the manuscript reading.
Response: As per your suggestion, it is well noted and considered. All the mentioned figures and tables are provided immediately after their mention in the text in the revised submitted manuscript.
Comment 7: L72-75: The authors should provide more detailed information about the combination of search terms for the articles. Which databases were consulted? What combinations of these words were used? What were the inclusion and exclusion criteria? If they do not have this information, I suggest removing it from the manuscript.
Response: Thanks. We removed it as per your professional suggestion for the quality of the manuscript.
Comment 8: I suggest removing the term "Results." To be considered a true result, a systematic search following guidelines such as PRISMA should have been conducted. As this was not done, it is a descriptive review and not systematic, so it is better to remove this term from the manuscript.
Response: Yes, as per your suggestion, we have removed the term “Result” from the manuscript since it is not a systematic review.
Comment 9: The authors should review and standardize the numbering of the manuscript's topics and subtopics, which are inconsistent.
Response: Thanks. We accept your comments and standardize all the numbers for topics and sub-topics in the revised manuscript highlighted with red color.
Comment 10: L92: Conditions of the mammary gland as a whole, not just the teats.
Response: L87: Yes, we rewrite it as mammary gland instead of teats
Comment 11: L100: Psychrotrophs as well as psychrophiles.
Response: L98: Thanks. We consider it.
Comment 12: L101: Reference the sources of contamination.
Response: L100: Thanks, reference (18) for the source of contamination is provided
Comment 13: In topic 1, I suggest including the following manuscript, which reports a technological problem not addressed in the review (https://doi.org/10.1016/j.idairyj.2021.105020).
Response: L157: In topic 3: Thanks, now it is included in the revised manuscript
Comment 14: L145: Spores can survive, so it cannot be considered true sterilization, only commercial.
Response: Thanks. Well noted
Comment 15: In topic 1, more details about the technological defects caused by the action of psychrotrophic enzymes should be described.
Response: L48-177: In topic 3: Thanks, we have updated and described more details in the manuscript as per the comment given
Comment 16: L176: Standardize the writing of the word Gram starting with a capital letter.
Response: L185: Thanks. Sorry, for the miss of the standard format while we wrote, we accepted and done.
Comment 17: L205, 215, 219, and other mentions in the manuscript: Spoilage instead of pathogen. The focus of the manuscript is on spoilage microorganisms, not pathogenic ones.
Response: L213, 220, 244: Thanks for your curious reviewing and we updated in the revised manuscript highlighted in red.
Comment 18: In the PCR topic, specify that the prior isolation of the microorganism in culture media is necessary.
Response: L259-261: Thanks and we specify it.
Comment 19: L344, 346: Typhimurium should be written without italics and starting with a capital letter, as it is a serovar, not a species of the Salmonella genus.
Response: L383, 385: Thank you very much. We have made the changes as per your comment.
Comment 20: L372: Replace the term microbes with microorganisms throughout the manuscript.
Response: L455: Thanks, we replace the term microbes throughout the manuscript with microorganisms
Comment 21: L391: Specify what "several days" would be. Be precise.
Response: L475: Thanks. We have specified the exact time needed a raw milk before processing in the edited manuscript.
Comment 22: In topic 5.1, the authors need to clarify which temperature control strategy should be used. It was difficult to extract practical and applicable information from the topic.
Response: L468: In sub-topic 6.1 on the revised manuscript: Sorry for lack of clarification, here what we need to clarify is that early combined treatment before processing with heating and then storage at < 10 °C refrigerator is more effective for both the mesophilic and psychrophilic spoilage count reduction.
Comment 23: L397: Wouldn't it be more applicable to solid products, like cheeses and butter?
Response:L481: Yes, it can be applicable for both liquid and solid products too. But we focus on raw milk spoilage reduction before processing.
Comment 24: In topic 5.3 on hygiene, the authors should include information on contamination sources in dairy farms and how milking and cleaning should be performed to reduce psychrotrophic counts.
Response: L499: In sub-topic 6.3 in the revised manuscript: Thanks, we do accept your valued comments, rewrite and we include the information needed in the revised manuscript.
Comment 25: In the topic on natural preservatives, the authors should provide more details and substances, with their mechanisms of action and practical applications.
Response: L560: In sub-topic 6.5 on the revised manuscript: Thanks for your valuable comments and we rewrite it in the revised manuscript.
Comment 26: L447: Do not abbreviate Pseudomonas as P.
Response: L593: Thanks we rewrite it
Comment 27: Adjust the Credit authorship according to the journal's standards.
Response: L637: Ok thanks, we adjusted the Credit authorship according to the journal standard in the rewritten submitted manuscript.
Thank you very much!!!
Reviewer 2 Report
Comments and Suggestions for Authors
The manuscript of Kidane Yalew et al [ foods-3159906 ] although it covers an interesting topic, it is frustratingly poorly prepared, which makes monitoring and reviewing extremely difficult.
1) please organize properly the numbering of sections, chapters, and subchapters
2) please improve the visibility of cited references in the references section because in the current form, it was difficult
3) English is confusing – I would recommend rewriting
4) Line 285- completely unclear which technique is more sensitive.
5) Line 292- ‘high light’- merge word
6) I can not see the point why figures 5th and 6th were submitted I would rather recommend the preparation of one big table with the pro and cons of all the mentioned techniques
7) The topic of cost-effectiveness was completely omitted it is sad
8) I would separate the section of methodology on how the references were selected
9) Was it validated or standardized how each publication was analyzed, and whether any tool (like CADIMA ) or manual cross-validation by co-authors was used for chosen references? No information in which databases manuscripts were screened.
10) I am not sure about the proper pathway of manuscript selection that pathway seems to be limited a lot.
11) I can not see the point of repetition of fragment Lines 370-383 I recommend rewriting that section
12) I have not found info about MS-based techniques that develop the most dynamic, especially for protein-based detection (the authors mainly refer to the undesirable effects of enzymes derived from bacteria). Strangely, there is no information about that. I recommend adding an additional section.
13) The lack of description for physical methods of milk preservation is also surprising (high pressure or ultrasound) are probably among the better described ones.
Comments on the Quality of English LanguageEnglish is not comfortable, but also not so bad. It could be easier so I recommend that rewriting should be done in some sections.
Author Response
A point-by-point response to reviewer
reviewer #2:
Comments and Suggestions for Authors
The manuscript of Kidane Yalew et al [foods-3159906] although it covers an interesting topic, it is frustratingly poorly prepared, which makes monitoring and reviewing extremely difficult. English is not comfortable, but also not so bad. It could be easier so I recommend that rewriting should be done in some sections.
Author response to reviewer #2
Thank you for your careful and thorough reading of this manuscript and for the comments and constructive suggestions, which help to improve the quality of our manuscript. The manuscript has now been modified having considered all comments and suggestions raised by the reviewers. We hope now that the revised manuscript is improved the English level and more readability. More work has been done as per your comments. A point-by-point response to the reviewers is provided below.
Comments 1: Please organize properly the numbering of sections, chapters, and subchapters.
Response: Thank you. As per your recommendation, we have properly put the number of sections, chapters and sub-chapters in the revised manuscript highlighted in red.
Comments 2: please improve the visibility of cited references in the references section because in the current form, it was difficult.
Response: Thank you we improve the visibility of cited references as per your comments in the revised manuscript.
Comments 3: English is confusing – I would recommend rewriting
Response: Thanks for your kindly advice. We rewrite the manuscript as per your comment in some sections for the quality of the manuscript in the revised manuscript.
Comments 4: Line 285- completely unclear which technique is more sensitive.
Response: L313-318: Sorry, you are right a bit unclear we rewrite it again, but the message we need to transfer is that LAMP is more sensitive than PCR method by two-fold.
Comments 5: Line 292- ‘high light’- merge word
Response: L322: Yes, it is a typing space error we take correction
Comments 6: I can not see the point why figures 5th and 6th were submitted I would rather recommend the preparation of one big table with the pro and cons of all the mentioned techniques
Response: L448: Table 3: Thank you. We remove the two figures and develop a table summarize the pro and cons all the mentiond techniques.
Comments 7: The topic of cost-effectiveness was completely omitted it is sad
Response: Yes, we didn’t find a published article that could show a monetary value comparison in terms of cost-effectiveness. Only we use the word cost effective as the techniques are mentioned in different published articles.
Comments 8: I would separate the section of methodology on how the references were selected
Response: Thank you. We just removed that part for the quality of the manuscript as per the first reviewer's suggestion.
Comments 9: Was it validated or standardized how each publication was analyzed, and whether any tool (like CADIMA ) or manual cross-validation by co-authors was used for chosen references? No information in which databases manuscripts were screened.
Response: Thanks for your kind advice. we accept your comment and removed
Comments 10: I am not sure about the proper pathway of manuscript selection that pathway seems to be limited a lot.
Response: Yes, thank you. we accept and remove that part for the quality of manuscript.
Comments 11: I can not see the point of repetition of fragment Lines 370-383 I recommend rewriting that section
Response: L452-467: Thank you. We rewrite that part and add some information too.
Comments 12: I have not found info about MS-based techniques that develop the most dynamic, especially for protein-based detection (the authors mainly refer to the undesirable effects of enzymes derived from bacteria). Strangely, there is no information about that. I recommend adding an additional section.
Response: L410: In section 5.1.7 in the revised. Thank you for your recommendation. We have included in the revised manuscript about MS.
Comments 13: The lack of description for physical methods of milk preservation is also surprising (high pressure or ultrasound) are probably among the better described ones.
Response: L529: in sub-topic 6.4 in the revised: Thank you and we add some description about the High pressure process physical preservation.
Thank you for your professional comments on the quality of the manuscript!
Reviewer 3 Report
Comments and Suggestions for Authors
Psychotrophic bacteria causing serious problems for dairy industry, so this manuscript covers important & actual topics, and advocating for sensitive methods of detection on early stages - on farm - of milk handling is valuable.
The abstract not fully represents the content of the manuscript, publications systematic analysis is not mentioned.
line 15 heat-resistant
line 17 "milking process" is not a good wording here
key words: the reviewer would add "cold storage" to the list as well
line 34 psychrophilic& psychrotrophic
line 48 hydrolitic
line 93 personel hygiene as well
Results 1. part contains lot of common knowledge dedicated to the enzymes - it can be removed without harm to the manuscript
Please check the numeration of Results 1, as we get few part numerated as 1
PCR methods - what about challenge with primers construction/strains mutation?
Part dedicated to methods of detection should contain more practical detection examples, as well material in which bacteria were detected - dairy products variety (or we are focusing on early stage of milk handling only?)
line 391 when we mention pasteurization we can not name the milk "raw" any more - according to the EU definitions
lines 421-422 GHP & GMP is a must as a baseline for HACCP in EU; part 5.3 must be more elaborated in details
line 434 bacteriocins
conclusions are too general
Tables are valuable addition to the manuscript. Figures legends are too brief. Fig. 1 is not necessary
Fig. 2 air&soil personal& barn& udder hygiene; the reviewer would change the factors line according to the milk chain
Fig. 5 microbiological milk analysis is not finished on CFU counts - if we would like to make microbs identification
Fig. 6 data from another publication, can not be present in this form - doubst about plagiarism
The manuscript needs more work to upgrade the presented knowledge and make it more detailed
Comments on the Quality of English LanguagePlease check the grammar & use of past tenses
some examples for correction:
line 64 grammar
lines 71-72, 76-78 need correction
fig. 7
Author Response
A point-by-point response to reviewer
reviewer #3:
Comments and Suggestions for Authors
Psychotrophic bacteria causing serious problems for dairy industry, so this manuscript covers important & actual topics, and advocating for sensitive methods of detection on early stages - on farm - of milk handling is valuable.
Author response to reviewer #3
Thank you for your careful and thorough reading of this manuscript and for the comments and constructive suggestions, which help to improve the quality of our manuscript. The manuscript has now been modified having considered all comments and suggestions raised by the reviewers. We hope now that the revised manuscript is improved the English level and more readability. More work has been done as per your comments. A point-by-point response to the reviewers is provided below.
Comments 1: The abstract not fully represents the content of the manuscript, publications systematic analysis is not mentioned.
Response: Thank you for your serious comment on the manuscript. The publication analysis was not included earlier in the abstract and now it is removed from the manuscript as per the comments forward by the two reviewers.
Comments 2: line 15 heat-resistant
Response: L16: Thank you. we rewrite it.
Comments 3: line 17 "milking process" is not a good wording here
Response: L18: Thank you. I accept your comment and rewrite it.
Comments 4: key words: the reviewer would add "cold storage" to the list as well
Response: L28: Thank you. We add it as per your comment.
Comments 5: line 34 psychrophilic& psychrotrophic
Response: L35-36:Thank you. We take your comment.
Comments 6: line 48 hydrolitic
Response: L49: Thank you. We rewrite it
Comments 7: line 93 personel hygiene as well
Response: L87: Thank you. We take your comment.
Comments 8: Results 1. part contains lot of common knowledge dedicated to the enzymes - it can be removed without harm to the manuscript
Response: Thank you. we have removed some common knowledge as per your comment and add some information in to it.
Comments 9: Please check the numeration of Results 1, as we get few part numerated as 1
Response: Thank you. We made corrections to all topics, sub-topics and sections in the revised manuscript as per your comment highlighted with red.
Comments 10: PCR methods - what about challenge with primers construction/strains mutation?
Response: Yes, you are right but as we compare with traditional is my message.
Comments 11: Part dedicated to methods of detection should contain more practical detection examples, as well material in which bacteria were detected - dairy products variety (or we are focusing on early stage of milk handling only?)
Response: Yes, you are right we are focusing on the raw milk. The comparison of different method has been listed in Table 2.
Comments 12: line 391 when we mention pasteurization we can not name the milk "raw" any more - according to the EU definitions
Response: L475: Thank you. We understand your point. We will rewrite it.
Comments 13: lines 421-422 GHP & GMP is a must as a baseline for HACCP in EU
Response: L521-525: We consider your concern
Comments 14: part 5.3 must be more elaborated in details
Response: L560: Part 6.5: Natural preservative in the revised manuscript. We added up some basic information as per your comment and the other reviewers comment too.
Comments 15: line 434 bacteriocins
Response: We corrected it.
Comments 16: conclusions are too general
Response: Thank you. We take your comment
Comments 17: Tables are valuable addition to the manuscript. Figures legends are too brief. Fig. 1 is not necessary
Response: Thank you. As per your comment and other reviewers comment we add table 3 and remove figure 1, 5 & 6.
Comments 18: Fig. 2 air&soil personal& barn& udder hygiene; the reviewer would change the factors line according to the milk chain
Response: Figure 1: In the revised manuscript: Thank you. We make changes as per your comment.
Comments 19: Fig. 5 microbiological milk analysis is not finished on CFU counts - if we would like to make microbs identification
Response: Thank you. We remove figure 5.
Comments 20: Fig. 6 data from another publication, can not be present in this form - doubst about plagiarism
Response: We remove figure 6 in the revised manuscript.
Comments 21: The manuscript needs more work to upgrade the presented knowledge and make it more detailed
Response: Thank you. We accept your valued comments
Comments on the Quality of English Language
Comments 22: Please check the grammar & use of past tenses
some examples for correction: line 64 grammar
Response: L66-68: Thank you. We have gone through grammar checking the whole revised manuscript.
Comments 23: lines 71-72, 76-78
Response: Thank you. Removed
Comments 24: need correction fig. 7
Response: Figure 3: In the revised manuscript, we have made some changes
Thank you for your professional comments on the quality of the manuscript!
Round 2
Reviewer 1 Report
Comments and Suggestions for Authors
All my suggestions were adressed.
Reviewer 3 Report
Comments and Suggestions for Authors
the manuscript shows serious improvement & its value is much higher, Authoers added important information and made the text more broad.
I accept current version, no further suggestions